# Physical Exercise Interventions Using Virtual Reality in Children and Adolescents with Cerebral Palsy: Systematic Review

**DOI:** 10.3390/healthcare13020189

**Published:** 2025-01-19

**Authors:** Javier Velasco Aguado, Mário C. Espada, Jesús Muñoz-Jiménez, Cátia C. Ferreira, Luisa Gámez-Calvo

**Affiliations:** 1Training Optimization and Sports Performance Research Group (GOERD), Faculty of Sport Science, University of Extremadura, 10005 Cáceres, Spain; jvelascow@alumnos.unex.es (J.V.A.); jsuliwan@unex.es (J.M.-J.); catia.ferreira@ese.ips.pt (C.C.F.); 2Instituto Politécnico de Setúbal, Escola Superior de Educação, 2914-504 Setúbal, Portugal; mario.espada@ese.ips.pt; 3Life Quality Research Centre (CIEQV), Instituto Politécnico de Setúbal, 2914-504 Setúbal, Portugal; 4Centre for the Study of Human Performance (CIPER), Faculdade de Motricidade Humana, Universidade de Lisboa, Cruz Quebrada-Dafundo, 1499-002 Lisboa, Portugal; 5Comprehensive Health Research Centre (CHRC), Universidade de Évora, 7004-516 Évora, Portugal; 6SPRINT Sport Physical Activity and Health Research & Innovation Center, 2001-904 Santarém, Portugal

**Keywords:** cerebral palsy, physical exercise, virtual reality, children, adolescents

## Abstract

Background/Objectives: Cerebral palsy (CP) is a neurological disorder that affects movement and posture. Physical activity (PA) is safe and crucial for healthy development; however, this population faces barriers that hinder its implementation. Virtual reality (VR) is an emerging and promising technology that promotes PA in young people with CP. This work aims to compile and analyze the current scientific literature on physical exercise (PE) programs using VR in children and adolescents with CP through a PRISMA systematic review. Methods: A systematic review was conducted and reported based on the PRISMA (Preferred Reporting Items for Systematic Review and Meta-analyses) statement. The search was conducted through the Web of Science, PubMed, and Scopus databases on 1st September 2024. Studies based on PA interventions using VR in children and adolescents with CP were selected. Results: A total of 24 experimental research articles were selected for this review. The studies included comprise a total sample of 616 participants between 4 and 18 years old. The studies involved a diverse range of interventions, from brief sessions to intensive training. The results consistently demonstrated improvements in motor control, muscle strength, aerobic capacity, and overall participation in daily activities. Conclusions: The results highlight that the use of VR for PE programs has numerous benefits such as increased enjoyment, facilitation of motor learning, and acquisition of functional skills. PE through VR in children and adolescents with CP represents a promising tool; more scientific and practical evidence is needed to confirm its long-term effectiveness.

## 1. Introduction

Cerebral palsy (CP) is the most common cause of motor disability in pediatric patients, with an estimated prevalence of 2 to 3 per 1000 live births [1]. It is defined as a permanent neurological disorder that primarily affects movement and posture and is characterized as non-progressive [2,3]. According to the National Institute of Neurological Disorders and Stroke (NINDS) [4], spastic CP, the most common type, is characterized by increased muscle stiffness and awkward movements, often affecting specific body parts; hemiplegic CP impacts one side of the body, such as an arm and leg on the same side; ataxic CP primarily affects balance and coordination, leading to shaky movements and difficulty with precise tasks; dyskinetic CP involves uncontrolled, involuntary movements, including twisting and repetitive motions; finally, mixed CP combines symptoms from more than one type, reflecting a combination of motor impairments. Individuals with CP present a wide range of motor symptoms, such as a lack of muscle coordination, excessive rigidity or flaccidity, tremors, or involuntary movements in addition to various non-motor manifestations, including epilepsy, sensory and cognitive deficits, language disorders, and sleep disturbances [5,6].

Physical exercise (PE) is safe and beneficial for individuals with CP [7], and specific recommendations for physical activity (PA) and exercise have been developed for this population [8]. PE can improve cardiorespiratory endurance and muscle strength and reduce sedentary behavior in individuals with CP [8]. However, several physiological, psychological, social, and macro-environmental factors act as barriers to the inclusion and participation of children and adolescents with CP in PA and/or sports [9,10]. Only 17.6% of individuals with CP meet PA recommendations, with 58% displaying sedentary behavior and 76.7% participating in PE for rehabilitation or treatment purposes, although interest in exercise tends to decline with age [11]. To counteract this, it is recommended to incorporate games and strategies to overcome these barriers [11].

Currently, virtual reality (VR) exercise has been recognized as a new method to promote PA [12]. The use of VR systems represents an innovative treatment approach that reinforces task-oriented motor learning and has a positive impact on the acquisition and improvement of functional skills [13], i.e., “the ability of an individual to perform daily activities that require physical effort” [14]. Additionally, VR allows for playful work on balance and postural control across different age ranges [15]. There are also low-cost VR devices that offer significant advantages for implementation, such as affordability, accessibility, technical support, easy updates with new technologies, and no need for additional modifications [13].

The use of VR as a tool for rehabilitation and PE in individuals with CP has shown significant benefits in terms of improving motor skills, coordination, and motivation during training. Recent studies have demonstrated that integrating VR into PE programs can increase adherence and enthusiasm for PA, as it provides an immersive and playful environment simulating various scenarios and challenges [16]. Moreover, VR allows exercises to be tailored to individual abilities, promoting a more personalized and effective experience [17]. These advancements suggest that VR has the potential to become a key tool in the rehabilitation of individuals with cerebral palsy, improving their quality of life through PA [18].

Therefore, the objective of this work is to conduct a systematic PRISMA review, as well as to compile and analyze the current scientific literature on the use of PE through VR in children and adolescents with CP. Furthermore, this work aims to determine and describe the potential effects of these interventions. It is expected that this study will provide information on the existing evidence related to the effects and possible applications of VR and PE interventions in individuals with CP.

## 2. Materials and Methods

### 2.1. Study Type and Design

This research constitutes a theoretical investigation [19], conducted through the compilation of scientific documents and the selection of studies [20]. A systematic literature review was conducted using the PRISMA method to identify and select relevant scientific studies (Appendix A) [21]. This review was registered in PROSPERO, an international database for systematic reviews, ensuring a transparent and standardized approach to the review process. The review protocol was prepared in advance and is publicly available, outlining the specific objectives, methodology, inclusion and exclusion criteria, and planned analysis. The ID for the PROSPERO registration is [CRD42024612250], providing readers with access to the detailed protocol for reference.

### 2.2. PICO Strategy and Inclusion and Exclusion Criteria

The PICO strategy (Population, Intervention, Comparison, and Outcome) was employed, helping in formulating the items that structure, specify, and articulate a series of questions based on the objectives of this research [22]. The questions posed in this review are as follows: What effects related to PE and VR can occur in the population with CP? What can this type of intervention offer compared to traditional therapy methods? And can it influence the participants’ predisposition to engage in PE? In this case, the items formulated through the PICO strategy help elucidate the opportunities offered by PE using VR for the CP population aged 5 to 17 years, as shown in Table 1.

Based on the PICO strategy and the research questions, a series of inclusion and exclusion criteria were implemented, as specified in Table 2.

### 2.3. Search Strategy

The search process was conducted in August 2024. The platforms used were Web of Science (WoS), Scopus (Elsevier), and PubMed (NIH). The search was performed in English, employing the following final search phrase (“virtual reality” AND “cerebral palsy” AND (“exercise” OR “sport*”)) in the same way across all databases, refining results by language (English and Spanish) and document type (Article).

### 2.4. Methodological Quality Analysis

The methodological quality of the studies was evaluated using the questionnaire developed by Law et al. [23]. The questionnaire consists of 16 Yes/No response questions.

The articles included in this review were assessed according to several criteria: the purpose of the study (Q1), the relevance of the background literature (Q2), the suitability of the study design (Q3), the study sample (Q4 and Q5), the use of informed consent (Q6), outcome measures (Q7 and Q8), description of the method (Q9), significance of the results (Q10), description of dropouts (Q13), practical consequences (Q15), and study limitations (Q16). Additionally, all articles were categorized into three methodological quality categories: The first category (C) includes those of low methodological quality, with a score equal to or less than 50%; the second category (B) includes those of good quality, scoring between 51% and 75%; and the third category (A) includes those of excellent quality, with a score above 75% [24].

The risk of bias in the selected studies was assessed using the Cochrane Risk of Bias tool (RoB) [25], a standardized method for assessing the risk of bias in randomized controlled trials and is increasingly being adapted for observational and quasi-experimental studies. It evaluates several domains, including selection bias, performance bias, detection bias, attrition bias, and reporting bias, which may threaten the internal validity of a study. Each domain is rated as having a low, high, or unclear risk of bias based on whether the study design and conduct have adequately addressed potential biases.

### 2.5. Variable Coding

The variables used were organized into two categories: general and specific variables, as well as variables related to methodological quality (Table 3).

### 2.6. Study Registration Procedure

This study followed a process and phases similar to those previously established in other studies [26,27], aimed at obtaining significant and relevant findings related to the research topic [28,29]. Figure 1 shows the flowchart of the search process. To determine whether a study met the inclusion criteria for this review, two authors independently screened each record and report retrieved. All records were organized in an Excel sheet, where the inclusion or exclusion decision was documented. Any disagreements between the two reviewers were resolved by consulting a third author, who independently reviewed the conflicting records. This process, shown in the Figure 2, ensured consistent and unbiased selection of studies for inclusion.

### 2.7. Synthesis Methods

The synthesis of results in this systematic review followed a narrative and descriptive approach, primarily integrating the findings from 24 studies. The narrative synthesis method was guided by the variability in study designs, interventions, and outcomes observed across the selected studies, which limited the feasibility of a meta-analytical approach. Furthermore, the data from eligible studies were categorized into general and specific variables, which facilitated the comparison and identification of recurring themes and patterns. Given the heterogeneity of interventions (use of various VR platforms like Nintendo Wii Fit, Xbox Kinect, and REAtouch^®^) and outcome measures (GMFM, muscle strength, balance, and engagement), a narrative synthesis allowed for a broader integration of qualitative and quantitative findings while acknowledging their contextual differences. Finally, results were presented in both tabular and textual formats, highlighting the intervention impacts, such as improved motor functions, balance, and engagement levels, across diverse settings and VR systems

## 3. Results

After applying the inclusion and exclusion criteria, the final sample consisted of 24 eligible studies on PE using VR in children and adolescents with CP. This section presents the variables analyzed in the different studies included in this review. Figure 3 shows the evolution in the number of studies related to PE using VR in children and adolescents with CP, published each year.

As shown in Table 4, the risk of bias assessment revealed several concerns across the studies reviewed. Most studies have an unclear risk of bias, as they do not clearly report the randomization process; furthermore, all studies show a high risk of performance bias and detection bias, as blinding of participants and personnel is generally not feasible for VR and rehabilitation interventions. However, in the attrition and reporting bias, all studies exhibit a low risk, as participant dropouts were minimal and well handled, and the outcomes were consistently reported as intended. These factors suggest that the findings of these studies should be interpreted with caution, as the presence of bias may affect the internal validity of their results.

Based on the publication years, the first experimental study was published in 2006, followed by a period of inactivity until 2011. From this year onward, except for 2019, studies were continuously published until 2023, with 2012 being the year with the highest number of studies, totaling five. Regarding the methodological quality of the studies, 21 of them received an A rating, and 3 received a B rating. The criteria most often unmet were criterion 5 (justification of the number of subjects) and criterion 13 (specification of the number of study dropouts).

Table 5 presents the results of the general variables and those related to the methodological quality of the studies included in the review. Additionally, Table 6 shows the results of the specific variables from the included studies. The studies are organized chronologically, from oldest to most recent, to facilitate reading.

The studies included in this review comprise a total sample of 616 participants (602 of them with CP], ranging from 4 to 18 years old, with the most common age group being between 6 and 12 years. In terms of gender, it is specified for 407 subjects, of whom 228 are boys and 179 are girls, showing gender parity among participants. Likewise, the main inclusion criteria mentioned in the reviewed studies are primarily gross motor function levels I to III on the GMFCS scale, along with the ability to follow simple instructions. Less frequently, but also used in several studies, were GMFCS levels I to III, a MAS score below 2, and the MUUL scale, the latter only in the study by Winkels et al. [36]. Regarding the types of CP studied, there is a wide heterogeneity in the sample, as several studies do not specify the type of CP [30,31,37,48]. Others only mention that spastic CP is studied [32,33,41,45], with hemiplegic CP being the most mentioned type [34,35,38,40,47,49,50,53]. On the other hand, ataxic CP is represented by a single subject [36] and dyskinetic CP by four subjects [42], among other types.

The studies analyzed different intervention programs using VR in children with CP across diverse goals, including improving motor function, balance, strength, hand–eye coordination, and overall functional mobility. VR platforms such as Nintendo Wii Fit, Wii Sports Resort, IREX, and custom systems were employed, with sessions ranging from short-term intensive programs to home-based interventions. Furthermore, Table 6 provides detailed and relevant information about the studies, including the variables and assessment tools used, the materials and types of VR employed, as well as the duration and type of exposure. It also outlines the specific measurement moments for the evaluated variables, offering a comprehensive overview of the methodologies and tools applied in the interventions. The duration and type of intervention have also been highly varied, ranging from a single session [30] to 12 weeks [32,33,42,47,50], and from mixed therapies dividing intervention time between some methodology and VR [30,38,39,40,41,42,45,47,51,52,53] to intensive PE training through VR lasting up to five and a half hours [49].

Among the most used types of VR in this review are the Wii [34,36,38,39,41,42,43,46,47], the Xbox Kinect [40,44,50,51,52], and the Interactive Rehabilitation and Exercise Systems—IREX [30,31,37,48].

## 4. Discussion

This PRISMA systematic review compiled and analyzed 24 studies related to exercise through VR in children and adolescents with CP. The studies included in this review comprise a total sample of 616 participants (602 of them with CP), ranging from 4 to 18 years old, showing gender parity among participants. Likewise, the main inclusion criteria mentioned in the reviewed studies are primarily gross motor function levels I to III on the GMFCS scale, along with the ability to follow simple instructions. This could be due to the widespread acceptance and standardization of the GMFCS, MACS, and MAS scales, in addition to the need for children to comprehend the games to perform them correctly and understand the feedback provided by the game itself.

Given the nature of the interventions in VR programs for children with CP, blinding and randomization were particularly challenging, which is common in this area of research. These factors are likely why many studies have a higher risk of bias in performance and detection domains. Regarding the methodological quality of the studies, 21 received an A rating, while 3 received a B rating. The criteria most frequently unmet were criterion 5, which concerns the justification for the number of subjects, and criterion 13, which relates to the specification of the number of study dropouts. These shortcomings suggest a need for more transparent reporting on sample size calculations and participant retention to strengthen the validity and reproducibility of the studies.

Regarding the types of CP studied, there is a wide heterogeneity in the sample; due to this heterogeneity, it is recommended that future studies with experimental designs include a more homogeneous sample and describe it in greater detail. Furthermore, according to the classification proposed by the NINDS [4], mixed CP has not yet been represented, and there is a very small sample size for tetraplegic, ataxic, and dyskinetic or athetoid CP. This may occur because 60–70% of people with CP present with spastic CP according to the Spanish Federation of Associations for the Assistance of People with Cerebral Palsy or Similar Disabilities [54]. Thus, it is recommended to conduct more studies on the benefits of using PE programs through VR in CP types with a limited sample size in this kind of research.

The duration and type of intervention have also been highly varied, ranging from a single session [30] to 12 weeks [32,33,42,47,50], and from mixed therapies dividing intervention time between some methodology and VR [30,38,39,40,41,42,45,47,51,52,53] to intensive PE training through VR lasting up to five and a half hours [49]. Although the results suggest a positive effect of including VR in various types of intervention, except in the case of Mills et al. [48], where the lack of benefit may be due to the short duration of the intervention, more rigorous studies are still needed to confirm these positive effects [12].

Among the most used types of VR in this review are the Wii [34,36,38,39,41,42,43,46,47], the Xbox Kinect [40,44,50,51,52], and the Interactive Rehabilitation and Exercise Systems—IREX [30,31,37,48]. Nintendo Wii_TM_ and IREX were mainly used in studies before 2019, while Kinect has been predominantly used in studies after 2015. Therefore, a trend shift towards Xbox Kinect can be observed in recent years. This may be due to its low cost and accessibility, as well as its continued innovation compared to the Wii, which is becoming more outdated. Furthermore, there is a clear predominance of low-cost VR types, as earlier recommended [13]. This will make PE practice through VR more affordable, facilitating its research and practical use by users.

The results indicate that PE programs through VR in children and adolescents with CP have shown repeated positive benefits in these interventions: improved movement control [30,39,40,41,51,52], balance [31,34,41,42,43,45,49,51], muscle strength [32,41,46,47,50], aerobic endurance [31,46,50], and participation in daily activities [36,42,53]. Additionally, increased interest, motivation, and fun are reflected [30,34,36,37,38,42,47], along with other benefits reported in fewer studies, such as improved bone mineral density [33] and reaction time [44]. These data demonstrate that PE in CP is essential to improving cardiorespiratory endurance and muscle strength and reducing sedentary behavior [8]. However, many factors, such as intensity, which ranges from moderate to intense [55], remain to be described, as well as studies that meet World Health Organization [56] recommendations. Therefore, it is recommended that future experimental articles include interventions that describe variables such as intensity in greater detail.

Most studies show greater benefits of VR compared to conventional training or therapy; however, in some cases, this difference is not seen [38,39], or it is shown in only half or fewer of the study variables [34,50,51]. This may be because VR-based PE is just an alternative or complement to conventional therapy but does not replace it. On the other hand, some of the benefits mentioned are not easily comparable between studies, such as gross motor function, which improved in the study of Gercek et al. [50] but did not in the study of Chen et al. [33], or agility, which improves in the study by Tarakci et al. [42] but not in the study by Jelsma et al. [34]. These discrepancies could be due to various factors, such as differences between groups, total intervention time, and the types of VR used, among others.

VR programs are feasible for school settings supervised by teachers, enabling implementation in rural areas where children may not receive continuous therapy [37], and in camps, improving motor function and occupational performance [49] and facilitating motor skill learning [53]. While VR integration presents multiple intervention settings, it also faces several barriers, such as the initial cost of equipment and the need for specialized staff training. Moreover, technological infrastructure in remote areas may be insufficient, hindering the effective implementation of these programs and limiting accessibility in contexts with a shortage of specialized human resources.

Regarding the maintenance periods of benefits after the interventions in different abilities, at least the following were maintained: 1 month for mobility [31], 2 months for balance [51], 3 months for movement quality, dexterity, and speed [35], 3 months for bone mineral density and lower limb muscle strength [32], transfer of learning to daily life [53], and 6 months for bimanual performance, unilateral function of the affected limb, and satisfaction with performance [49]. Although these results are limited, they show that the benefits are durable. This may be because task-oriented learning benefits functional skills [13], allowing the improvement of these skills to be utilized in daily life [14]. Therefore, it is recommended that future studies describe this benefit maintenance period more thoroughly.

Among the strengths of this work is the use of PRISMA methodology, which ensures a comprehensive search and rigorous selection of relevant studies, providing a complete and well-structured view of the topic. Additionally, the focus on a specific population, such as children and adolescents with CP, and the use of innovative technology like VR, highlights the relevance and potential impact of this study in improving therapeutic interventions. However, this work faces some limitations, such as the possible heterogeneity of the included studies in terms of design, sample size, and outcome variables. Finally, the rapid technological evolution implies that the conclusions of this review could become outdated in a short time, underscoring the need for continuous and updated research in the future.

## 5. Conclusions

PE using VR in children and adolescents with CP shows significant benefits. It offers an accessible and affordable alternative with low-cost devices, enhancing motivation and interest in PA in this population. Additionally, it facilitates motor learning and the acquisition of functional skills. However, further research is needed in this field to optimize interventions and develop this tool, thus overcoming barriers that limit the participation of this population in PA.

## Figures and Tables

**Figure 1 healthcare-13-00189-f001:**
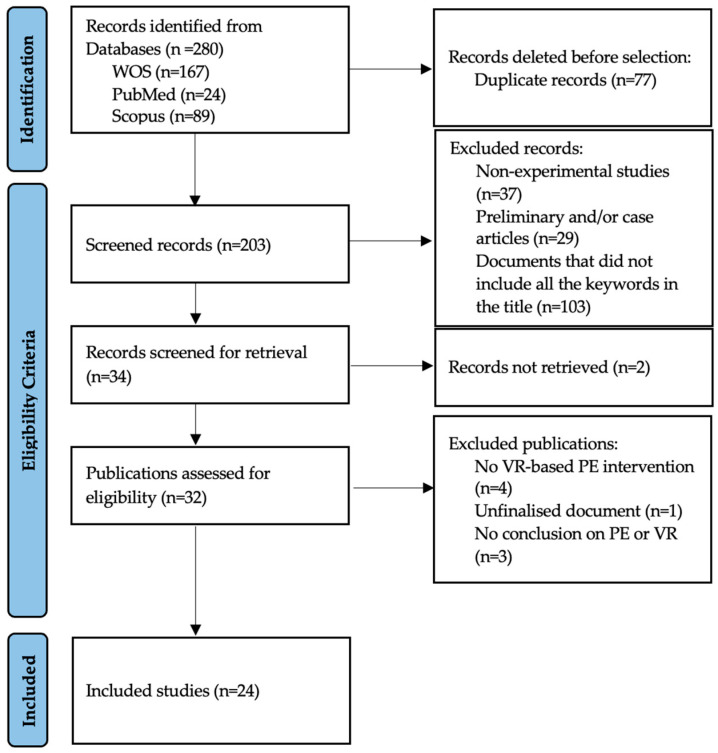
PRISMA flow diagram.

**Figure 2 healthcare-13-00189-f002:**
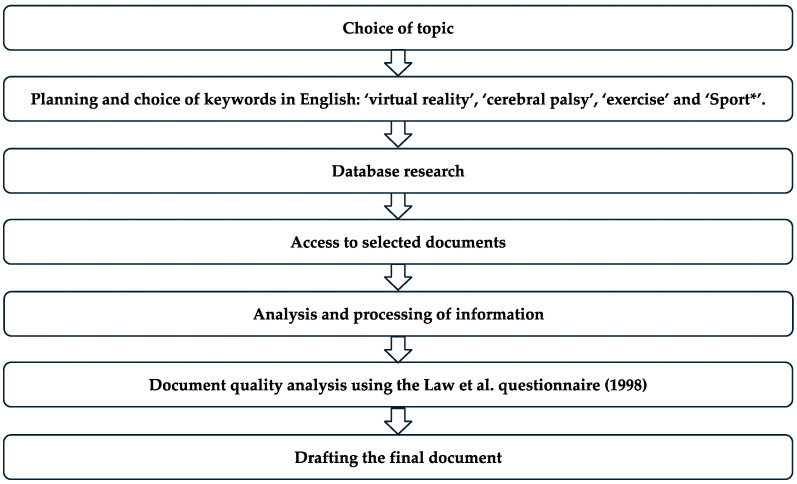
Study phases (own elaboration scheme).

**Figure 3 healthcare-13-00189-f003:**
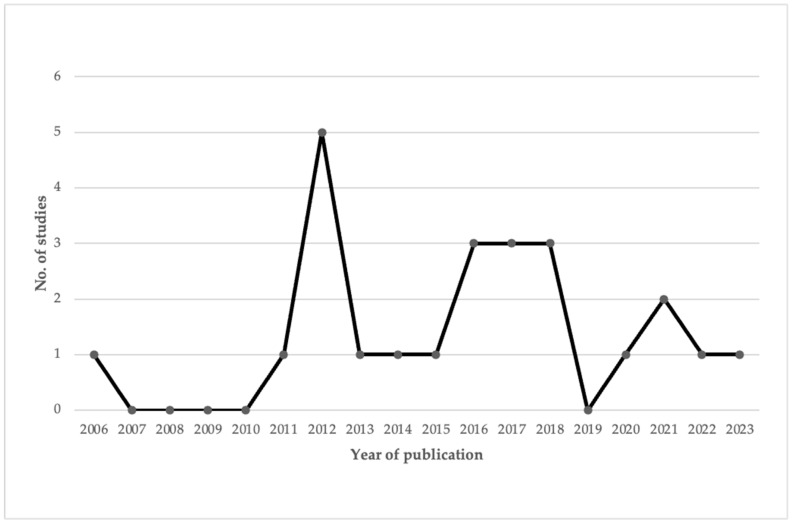
Studies and year of publication.

**Table 1 healthcare-13-00189-t001:** PICO strategy related to PE, VR, and CP.

Population	Intervention	Comparison	Outcomes
People with CP aged 5 to 18 years	PE program using VR as the main tool	Traditional intervention methods or no therapy	Provides results related to quality of life and willing to engage in PE

PE, physical exercise; VR, virtual reality; CP, Cerebral palsy.

**Table 2 healthcare-13-00189-t002:** Inclusion and exclusion criteria.

Selection Criteria	Inclusion
Document Type	Scientific article.
Study Type	Documents with experimental design.
Study Subjects	Sample of subjects aged 5 to 18 years with CP.
Methodology	Intervention focused on a PE program using VR.
	**Exclusion**
Study Type	Documents with non-experimental methodologies, case studies, or preliminary studies.
Medical Conditions	Studies include participants with additional medical conditions that may affect intervention participation.
Conclusion	Conclusions that focus on aspects other than PE and VR or do not report intervention outcomes.

PE, physical exercise; VR, virtual reality; CP, Cerebral palsy.

**Table 3 healthcare-13-00189-t003:** Explanation of the variables used in this review.

**General Variables**	
Author/s	Original author/s of the publication.
Year	Year of publication.
Sample	Number of participants, age, gender, and type of CP.
Objective	Main objectives of each study.
Conclusions	Key conclusions of the study.
**Specific Variables**	
Study variables and their evaluation tools	Variables measured in each study and the various tools used.
Material and type of VR	Materials and used VR tools.
Type and duration of exposure and measurements	Specifies the type and duration of the intervention performed and the measurements taken.
Main results	Results from the measurements obtained from the study variables.
**Methodological Quality**	
Methodological quality	The Law et al. [23] questionnaire was applied to the studies, with the rating proposed by Sarmento et al. [24].

VR, virtual reality; CP, Cerebral palsy.

**Table 4 healthcare-13-00189-t004:** Risk of bias of included studies.

Study and Publication Year	Selection Bias	Performance Bias	Detection Bias	Attrition Bias	Reporting Bias	Other Bias
Bryanton et al., 2006 [30]	Unclear	High	High	Low	Low	Unclear
Brien and Sveistrup, 2011 [31]	Unclear	High	High	Low	Low	Unclear
Chen et al., 2012a [32]	Low	High	High	Low	Low	Unclear
Chen et al., 2012b [33]	Low	High	High	Low	Low	Unclear
Jelsma et al., 2012 [34]	Unclear	High	High	Low	Low	Unclear
Rostami et al., 2012 [35]	Low	High	High	Low	Low	Unclear
Winkels et al., 2013 [36]	Unclear	High	High	Low	Low	Unclear
Rosie et al., 2013 [37]	Unclear	High	High	Low	Low	Unclear
Chiu et al., 2014 [38]	Low	High	High	Low	Low	Unclear
Shin et al., 2015 [39]	Unclear	High	High	Low	Low	Unclear
Bedair et al., 2016 [40]	Low	High	High	Low	Low	Unclear
Cho et al., 2016 [41]	Low	High	High	Low	Low	Unclear
Tarakci et al., 2016 [42]	Low	High	High	Low	Low	Unclear
Gatica-Rojas et al., 2017 [43]	Low	High	High	Low	Low	Unclear
Pourazar et al., 2017 [44]	Unclear	High	High	Low	Low	Unclear
Yoo et al., 2017 [45]	Unclear	High	High	Low	Low	Unclear
Chiu et al., 2018 [46]	Low	High	High	Low	Low	Unclear
El-Shamy et al., 2018 [47]	Unclear	High	High	Low	Low	Unclear
Mills et al., 2018 [48]	Low	High	High	Low	Low	Unclear
Roberts et al., 2020 [49]	Low	High	High	Low	Low	Unclear
Gercek et al., 2021 [50]	Low	High	High	Low	Low	Unclear
Jha et al., 2021 [51]	Low	High	High	Low	Low	Unclear
Hamed et al., 2022 [52]	Unclear	High	High	Low	Low	Unclear
Saussez et al. 2023 [53]	Unclear	High	High	Low	Low	Unclear

Notes: Id.: Identification.

**Table 5 healthcare-13-00189-t005:** General and methodological quality variables of the studies.

ID	Citation and Publication Year	Sample	Objectives	Conclusions	Q
1	Bryanton et al., 2006 [30]	16 children aged 7 to 17 years, 10 with CP level I-II of GMFCS (4 boys and 6 girls), and 6 without CP (2 boys and 4 girls). Subjects with CP had a GMFCS level I-II.	Compare the effectiveness of motor control exercises in children with CP using VR or conventional methods, assessing children’s motivation and compliance and whether improvements in ROM and hold time transfer to daily activities.	VR exercises for children with CP resulted in greater ankle ROM, better movement control, and increased interest compared to conventional exercises.	B
2	Brien & Sveistrup, 2011 [31]	4 adolescent boys aged 14 to 18 years, with CP, level I of GMFCS, and the ability to follow standardized test instructions.	Conduct a short-term intensive VR program aimed at improving balance and mobility in adolescents with CP.	High-level balance skills and functional mobility are modifiable in this population.	A
3	Chen et al., 2012a [32]	27 children, aged 6 to 12 years, both genders, with spastic CP levels I-II of GMFCS.	Analyze an HVCT program and its impact on ABMD in ambulatory children with CP.	The muscle-strengthening program is more specific for improving bone density in children with CP than general physical activity. The proposed protocol is an effective and efficient strategy to improve ABMD in the lower limbs.	A
4	Chen et al., 2012b [33]	28 children, aged 6 to 12 years, both genders, spastic CP levels I-II of GMFCS.	Evaluate the effect of an HVCT program to improve muscle strength in ambulatory children with spastic CP.	The HVCT program did not improve gross motor function but significantly increased knee muscle strength in children with spastic CP. The greatest benefits were seen in knee flexor muscles.	A
5	Jelsma et al., 2012 [34]	14 children, aged between 7 and 14 years, 8 boys and 6 girls, with spastic hemiplegic CP and GMFCS levels I-II.	The aim was to evaluate the impact of training with Nintendo Wii Fit on balance and gross motor function in children with spastic hemiplegic CP and compare it to conventional physiotherapy.	Training with Nintendo Wii Fit improved balance in children with spastic hemiplegic CP, and there was a preference for VR training over conventional physiotherapy. However, there were no significant improvements in agility or the ability to go up and down stairs. It was recommended as a useful complement to conventional physiotherapy but not as a replacement.	A
6	Rostami et al., 2012 [35]	32 participants, aged 6 to 11 years, 18 girls and 14 boys, with spastic hemiparetic CP with a MAS score below 3.	Determine the effects of a modified constraint-induced movement therapy practice period in a virtual environment on upper limb function in children with spastic hemiparetic CP.	Modified constraint-induced movement therapy in a virtual environment could be a promising rehabilitation procedure to enhance the benefits of both VR and this type of therapy.	A
7	Winkels et al., 2013 [36]	15 children, aged between 6 and 15 years, 12 boys and 3 girls, with CP (8 with bilateral spastic CP, 6 with unilateral spastic CP, and 1 with ataxic CP) and the ability to hold the Wii controller, with a minimum score of 11% on the MUUL.	To explore the effects of Nintendo Wii training on the upper limbs in children with CP, as well as to evaluate user satisfaction and usability for both participants and professionals and to assess if the intervention was enjoyable.	The use of VR and games can be very interesting as a tool, and it can be motivating and help improve functional arm performance in children with CP. However, not all games may serve this purpose, so specially developed or adapted games would be a better option.	A
8	Rosie et al., 2013 [37]	5 children, aged 7 to 13 years, 3 girls and 2 boys, with CP, level I-II in GMFCS.	Evaluate the feasibility of a school-based virtual rehabilitation program for children with CP and whether it is feasible to implement the IREX system supervised by teachers in isolated areas without regular access to therapy.	The IREX system is feasible to implement in a school environment supervised by teachers. It is an option to provide physical therapy to children in isolated areas who do not receive continuous therapy.	B
9	Chiu et al., 2014 [38]	62 participants, aged between 6 and 13 years, 28 boys and 34 girls, with spastic hemiplegic CP and sufficient manual function to hold the Wii remote control.	To evaluate the effectiveness of Wii Sports Resort training in improving hand coordination, strength, and function in children with hemiplegic CP to see if improvements are sustained up to 6 weeks post-intervention and to assess the feasibility of home-based training.	There were no significant improvements between the two groups, but caregivers reported increased hand use after Wii Sports Resort training. Commercial video games could be a motivating form of home exercise for children with CP.	A
10	Shin et al., 2015 [39]	16 children, aged 4 to 8 years, 9 boys and 7 girls with spastic diplegic CP, with motor function level I, II, or III on the GMFCS scale.	Evaluate the effects of conventional neurological treatment and a virtual reality training program on hand-eye coordination in children with CP.	A well-designed VR training program can improve hand-eye coordination in children with CP.	A
11	Bedair et al., 2016 [40]	40 children, aged 5 to 10 years, 23 boys and 17 girls, with spastic hemiplegic CP and a grade 1+ or 2 on the MAS.	To assess the effects of VR games as a complementary tool for upper limb treatment in children with spastic hemiplegia.	VR games significantly improved motor and visual skills in children with spastic hemiplegia. It is believed that the children’s active participation and motivation in a simulated environment contributed to these improvements, facilitating cortical reorganization and the development of new motor pathways.	A
12	Cho et al., 2016 [41]	18 children, aged 4 to 16 years with spastic CP, level I-III on GMFCS, and a score of less than 2 on MAS.	Investigate the effects of VR treadmill training [VRTT) in children with CP, comparing it to traditional treadmill training, focusing on improving gait, balance, muscle strength, and gross motor function.	VRTT is effective in improving gait, balance, muscle strength, and gross motor function in children with CP, although future studies are suggested to confirm long-term efficacy.	A
13	Tarakci et al., 2016 [42]	30 children, aged 5 to 18 years, 19 boys and 11 girls, 12 with diparetic CP, 14 with hemiparetic CP, and 4 with dyskinetic CP, with levels I-III on the GMFCS.	To determine if Wii Fit video games improve static and dynamic balance and independence in daily activities compared to conventional balance training in children with mild CP.	Nintendo Wii Fit training combined with NDT significantly improved static and dynamic balance in children with mild CP. It also resulted in greater independence in daily activities, and increased motivation and satisfaction among both children and their families, making the treatment more appealing.	A
14	Gatica-Rojas et al., 2017 [43]	32 children, aged 7 to 14 years, gender not specified, with spastic hemiplegic and diplegic CP, with levels I-II on the GMFCS.	To compare the post-treatment effects and effectiveness of Nintendo Wii Balance Board therapy versus standard physiotherapy on standing balance in children and adolescents with CP.	Nintendo Wii Balance Board therapy improved balance in CP patients more than standard physiotherapy, especially in those with spastic hemiplegia. However, the positive effects diminished by the 2nd and 4th week after the intervention. Wii therapy is easy to implement in rehabilitation centers and can be a useful tool to improve balance in CP patients.	A
15	Pourazar et al., 2017 [44]	30 boys, aged 7 to 12 years, all with spastic hemiplegic CP, level I-III on GMFCS, and level I-II on MACS.	Investigate the effects of VR intervention program training on reaction time in children with CP.	This study suggests VR as a promising tool in the rehabilitation process to improve reaction time in children with CP.	A
16	Yoo et al., 2017 [45]	18 children (2 girls and 10 boys with spastic CP and 3 girls and 5 boys without CP), aged 9.5 ± 1.96 years for CP children, and 9.75 ± 2.55-year-old children with CP had a level I-III on the MACS and a grade 1 on the MAS.	Compare the therapeutic effects of VR-augmented EMG biofeedback and EMG biofeedback alone on the imbalance of tricep and bicep muscle activity and coordination of elbow joint movement during a reaching motor task.	VR-augmented EMG feedback produced better neuromuscular control of the elbow joint than EMG feedback alone.	A
17	Chiu et al., 2018 [46]	20 children, aged 6 to 12 years, 11 boys and 9 girls, with CP (10 with diplegia, 8 with hemiplegia, and 2 with tetraplegia), and with levels I-III on the GMFCS.	To assess whether Wii Fit training is feasible and can improve strength, balance, mobility, and participation in children with CP.	Home-based Wii Fit training is feasible and safe for children with CP and appears to have clinical benefits for strength and mobility in this population. A randomized controlled trial is suggested to investigate further.	A
18	El-Shamy et al., 2018 [47]	40 children, aged 8 to 12 years, 26 boys and 14 girls, with spastic hemiplegic CP, capable of following simple instructions, with no prior Wii experience, and able to use the Wii safely, with levels I-III on the MACS.	To investigate the effect of Nintendo Wii training on the function of the affected hand in children with hemiplegic CP.	Wii training, combined with standard therapy, can reduce spasticity, increase grip strength, and improve manual function in children with spastic hemiplegic CP. The intervention had a high adherence rate, suggesting that Wii games can be a motivating and effective tool in rehabilitation.	A
19	Mills et al., 2018 [48]	11 children, aged 7 to 17 years, 6 boys and 5 girls, all with CP and a level I-II of GMFCS.	Evaluate the effects of a 5-day VR-based exercise program on anticipatory and reactive postural control mechanisms in children and youth with CP.	No effect was observed from the 5-day VR-based intervention on postural control mechanisms used in response to platform perturbations.	B
20	Roberts et al., 2020 [49]	31 children (16 boys and 15 girls) with hemiplegic CP, aged 5 to 15 years, with a classification of I-III on the MACS.	Determine the acceptability and effects of a P-CIMT camp for children with hemiplegic CP augmented by using an exoskeleton for playing in VR.	A P-CIMT camp augmented by the Hocoma Armeo Spring Pediatric was feasible and accepted by participants. Bimanual hand function and occupational performance improved immediately after the intervention, and treatment effects persisted for at least 6 months.	A
21	Gercek et al., 2021 [50]	19 children, aged 6 to 12 years, 14 boys and 5 girls, with hemiplegic CP, and level I and II on GMFCS.	Investigate how virtual and traditional golf affects balance, muscle strength, lower limb flexibility, and aerobic endurance in children with CP.	Both virtual and traditional golf training can be effective complementary applications for improving lower limb functions and physical performance in children with CP.	A
22	Jha et al., 2021 [51]	38 children, aged 6 to 12 years, all with bilateral spastic CP with a GMFCS level of II-III and I-III on the MACS.	Examine the effects of VR games and physiotherapy on balance, gross motor performance, and daily functioning in children with bilateral spastic CP.	The combination of VR games and physiotherapy is not superior to physiotherapy alone in improving gross motor performance and daily functioning but is better for balance in children with bilateral spastic CP.	A
23	Hamed et al., 2022 [52]	30 children (boys and girls) aged 7 to 10 years with spastic diplegic CP with a score of 2 or less on MAS.	Examine the effects of VR games on motor performance levels in children with spastic CP.	The use of a VR-based video game system in the rehabilitation program for patients with CP could be considered an effective intervention method and could be added to treatment programs, as it resulted in a significant improvement in motor performance levels.	A
24	Saussez et al. 2023 [53]	40 children, aged between 5 and 18 years old, 20 boys and 20 girls, all with hemiplegic CP with levels I-II on the GMFCS and I-III on the MACS.	To compare whether the integration of REAtouch^®^ in the HABIT-ILE intervention is at least as effective as the conventional HABIT-ILE intervention for children with hemiplegic CP.	The use of the virtual REAtouch^®^ device is not inferior in efficacy compared to conventional HABIT-ILE intervention in children with CP. This demonstrates the feasibility of using this device and establishes the possibility of applying therapeutic principles of motor skills learning in sessions based on virtual environments.	A

ABMD: Areal Bone Mineral Density; EMG: Electromyography; GMFCS: Gross Motor Function Classification System; HABIT-ILE: Hand–Arm Bimanual Intensive Therapy Including Lower Extremities; HVCT: Home-Based Virtual Cycling Training; ID: Identifier; IREX: Interactive Rehabilitation and Exercise Systems; MACS: Manual Ability Classification Scale; MAS: Modified Ashworth Scale; MUUL: Melbourne Assessment of Uni-lateral Hand Function; NDT: Neuro-Developmental Treatment; P-CIMT: Pediatric Constraint-Induced Movement Therapy; Q: methodological quality; REATouch^®^: Augmented Reality and Tactile Technology; ROM: range of motion; VRTT: Virtual Reality Treadmill Training.

**Table 6 healthcare-13-00189-t006:** Specific variables.

ID	Variables and Assessment Tools	Material and Type of VR	Type and Duration of Exposure and Measurement Moments of Variables	Main Results
1	Selective motor control and exercise compliance were both variables measured using an electrogoniometer that recorded ankle ROM.	IREX VR system, including a large TV monitor, a camera, and a computer. Two VR applications, Coconut Shooters and Ninja Flip, were created to introduce ankle movements into the virtual environment.	A single 90 min exercise session, alternating 10 min blocks between conventional and VR exercises. Exercises were performed seated on the floor and in a chair, aiming to dorsiflex the ankle to the maximum range, hold for 3 s, relax, and repeat. The measurements were taken during the training.	Children showed more interest and fun with VR games. Conventional exercises completed more repetitions at the same time, but VR exercises achieved greater ROM and retention time in the extended position.
2	Balance and Community Mobility Scale (CB and M), 6-Minute Walk Test (6MWT), Timed Up and Down Stairs (TUDS), and Gross Motor Function Measure Dimension E (GMFM-E).	Commercial VR system with a 32-inch screen, a video camera, and a high-performance computer. The IREX software version 1.4 was used to create a balance training program with applications like Soccer, Zebra Crossing, Snowboard, and Gravball.	90 min of VR-based balance training for 5 consecutive days, divided into two 45 min sessions with a 30 min break in between. Measurements were taken between 3 and 6 evaluations during the week prior to the intervention, before the training sessions on days 2 to 5, and 3 times the week after, as well as 1 month after the intervention.	Statistically significant improvements were observed in all tests (CB and M, 6MWT, TUDS, and GMFM-Dimension E). Improvements persisted for at least 1-month post-intervention.
3	Muscle strength (curl-up scores and isokinetic torques of knee extensors and flexors measured with an isokinetic dynamometer), lumbar spine and distal femur ABMD (DXA), and motor function (GMFM-66).	Eloton SimCycle Virtual Cycling System connected to a personal computer using Eloton Theater CD-ROMs, which allowed access to interactive virtual worlds. The cycling resistance was adjusted using a nylon tension strap.	12-week intervention with 3 sessions of 40 min per week. The control group (14 children) completed general physical activities at home, and the experimental group (13 children) performed HVCT. Measurements were taken between 3 and 6 evaluations during the week prior to the intervention, before the training sessions on days 2 to 5, and 3 times the week after, as well as 1 month after the intervention.	The HVCT group showed significant improvement in distal femur ABMD, and knee extensor and flexor strength compared to the control group. No significant differences were found in lumbar spine ABMD between groups, and no significant differences were observed in GMFM-66.
4	Isokinetic knee extensor and flexor torque (isokinetic dynamometer), gross motor function (BOTMP and GMFM-66), and strength change index.	Eloton SimCycle Virtual Cycling System and VR environment connected to a computer and CD-ROMs guiding users through virtual exercises.	RCT with two random groups: an experimental group and a control group. The intervention lasted 12 weeks, with the experimental group performing HVCT 3 times a week for 40 min, including warm-up, cycling, and cool-down. Measurements were taken before and after the 12 weeks of intervention.	HVCT significantly improved muscle strength without affecting gross motor function. Flexor muscles of the knee showed greater strength increases than extensors.
5	Balance and RSA (BOTMP-2) were measured, as well as functional mobility (TUDS).	The Nintendo Wii Fit was used, which included a balance board that measured the user’s weight distribution and center of pressure. The Wii Fit games played were snowboarding, skiing, and hula hoop.	Participants were randomly assigned to two groups; one group started in week 3 and the other in week 5. The intervention consisted of 25 min sessions with the Nintendo Wii Fit four times a week for three weeks. The measurements of the variables were taken before, during, and weekly up to 9 weeks after the intervention.	There was a significant improvement in balance, but no improvements were observed in agility, speed, or the ability to ascend and descend stairs. Of the 14 children, 10 preferred the intervention with the Wii, but the improvement did not transfer to overall function.
6	Variables such as the use of the affected limb and quality of movement (QOM) were measured using the PMAL, while voluntary displacement (SD) was assessed using the BOTMP.	A Human–Machine interface was employed with the E-Link Upper Limb Exerciser (E 3000), and the E-Link Evaluation and Exercise System was used as a VR tool. Various virtual games in this system were used to be engaging and motivating from the participants’ perspective.	The group was divided into four subgroups: one completed only CIMT, another only VR, one combined both, and a control group. Over four weeks, each group completed 3 sessions per week, 90 min each, with measurements taken before, after, and three months post-treatment.	The group combining modified CIMT and VR showed significant improvements in limb use, MC, and VD, which were maintained during the 3-month follow-up. The VR and modified CIMT groups also improved but not as much as the combined group, and the control group showed no significant improvements.
7	The quality of upper limb movements was assessed (MAUULF), along with the ease with which participants performed daily activities (ABILHAND-Kids scale), and questionnaires on user satisfaction and usability for healthcare professionals were administered.	The Nintendo Wii video game console was used as a form of virtual reality, employing sports games like boxing and tennis.	A 6-week intervention was conducted, with training sessions on the Nintendo Wii twice a week for 30 min, during which boxing, and tennis were played using the more affected arm. Measurements were taken before and after the intervention.	There were no significant changes in the quality of upper limb movements; however, a significant improvement in the ease of performing daily activities was observed, along with satisfaction from both children and healthcare professionals, and enjoyment from the children.
8	Children’s motivation to participate in the IREX, their perception of the games (both measured through a structured questionnaire), and the feasibility of implementing IREX in a school setting supervised by teachers (measured through a qualitative interview) were evaluated.	The VR material used was IREX, developed by Gesture Tek, USA. This system provided a virtual sports or gaming environment that promoted motivation despite the repetitive nature of therapy, tailored to everyone.	Each participant received an individualized IREX program, which they followed for 8 weeks, 3 times a week for 30 min. The IREX was placed in the child’s school during the intervention and was supervised by a teaching assistant. The measurements were taken before and after the intervention.	The children found IREX fun and easy to use, improvements in arm movement were reported, and there was a desire to continue using IREX. The supervisor played a crucial role in the success of the intervention, and it was feasible and well-received by the teaching assistants.
9	The measurements included coordination (tracking task on a screen), strength (Power Track II dynamometer), hand function (NPT and JTTHF), and caregivers’ perceptions of hand function (Functional Use Survey).	The Wii Sports Resort game was used as a virtual reality tool, and the games selected were Bowling, Air Sports, Frisbee, and Basketball, chosen for their ability to work on upper extremities and provide immediate feedback.	Participants were divided into an experimental group (32 children) who underwent 6 weeks of training with Wii Sports Resort alongside their usual therapy and a control group (30 children) that received only the usual therapy. Measurements were taken before, immediately after, and 6 weeks after the intervention by a blinded evaluator.	There were no significant differences in coordination and hand function between the two groups. Regarding grip strength, there was a trend toward improvement in the experimental group, although it was not statistically significant. Caregivers reported a greater use of the hand in the experimental group after 6 weeks post-intervention.
10	Eye–hand coordination (EHC) and visual–motor speed (VMS) were measured.	A series of conventional therapeutic exercises were performed, and the VR group used a program with the Nintendo Wii as a tool to foster interest and participation in children.	The study lasted 8 weeks, with the control group doing 45 min of therapeutic exercise twice a week, while the VR group conducted 30 min of therapeutic exercise and 15 min of VR training twice a week. The measurements were taken before and after the intervention.	Both groups showed significant improvements in EHC and VMS, but there were no significant differences between the groups.
11	Object manipulation and visual-motor skills (PDMS-2) and manual function (ABILHAND-Kids) were measured.	The Xbox Kinect was used, and games played included tennis, bowling, golf, space pop, bubbles, and boat driving.	The group was divided into two groups of 20 participants each. The study group used the Kinect for 30 min three times a week, in addition to 60 min of physical therapy, while the control group only underwent 60 min of therapy. This intervention lasted for four months, with measurements taken before, midway through, and at the end of the intervention.	Both groups showed significant improvements in all study variables, but the control group demonstrated significantly greater improvements.
12	Muscle strength (measured with a digital manual muscle tester), gross motor function (GMFM), balance (PBS), walking speed (10MWT), and walking endurance (2MWT) were assessed.	The Nintendo Wii Fit Plus program was used for the VRTT, with participants walking with a Nintendo Wii remote at their waist, which recorded body movement acceleration using a 3D accelerometer and transmitted the information to a 42-inch television.	Participants were divided into two groups of 9, one performing VRTT and the other traditional treadmill training exercises. Both groups performed 30 min of exercise, 3 days a week for 8 weeks, along with general physical therapy for 30 min, 3 times a week for 8 weeks. Measurements were taken before and after the intervention.	Participants in the VRTT group showed significant improvements in all areas compared to the traditional exercise group.
13	Measurements included: balance (FFRT, FSRT, Nintendo Wii Fit balance, and game scores), agility (STST and TGGT), walking speed (10MWT), stair climbing (10ST), and functional independence (WeeFIM).	The Nintendo Wii console and balance board were used, along with Wii Fit games, which included ski slalom, tightrope walking, and heading a ball. All activities were projected onto a screen in a dark room for the best immersive experience.	Participants were randomly assigned to either the control group, which received NDT and conventional balance training, or the experimental group, which received NDT and played balance video games on the Wii Fit console. A total of 24 interventions were conducted over 12 weeks, with measurements taken before and after the intervention.	Both groups showed significant improvements across all variables, and the experimental group demonstrated significant improvements in balance, agility, and functional independence compared to the control group.
14	The area of sway of the center of pressure was evaluated using the AMTI OR6-7 force platform, along with standard deviation in the medial–lateral and anteroposterior directions, and the velocity of the center of pressure in these same directions (AMTI NetForce software was used for data collection, and MATLAB R2012 software was utilized for processing and calculating variables).	The Nintendo Wii Balance Board was used as virtual reality material, and a systematic exercise program called Wii therapy was employed, with games including Snowboard, Penguin Slide, Super Hula Hoop, Run Plus, and Heading Football.	Participants were divided into two groups: the experimental group, which received therapy with the Nintendo Wii balance board, and the control group, which received standard physiotherapy, with both groups consisting of 16 participants. Both groups underwent 6 weeks of intervention with 3 therapy sessions per week. Measurements were taken before the intervention and every two weeks from the start of the study until week 10 (4 weeks after the completion of the intervention).	Compared to standard physiotherapy, the experimental group significantly reduced the sway area and anteroposterior deviation in the eyes-open condition. The positive effects associated with Wii therapy diminished between weeks 8 and 10, with only children with spastic hemiplegia showing significant improvements with Wii therapy, and all participants completed the therapies without adverse effects.
15	Reaction time (SRT and DRT, both measured by the RT-888 device) and general health (GHQ) were measured.	The RV intervention device used was the Xbox 360 Kinect, and bowling (Brunswick Pro-Bowling) and golf (The Golf Club) games were utilized.	Participants were randomly divided into an experimental and control group. The experimental group participated in a 12-session RV intervention program over 4 weeks to improve motor skills and reaction time. Measurements were taken before and one day after the intervention.	After the 4-week RV intervention program, the experimental group showed a significant improvement in reaction time compared to the control group.
16	Muscle activity (EMG), movement coordination during a reaching task (3-axis accelerometer), elbow extension range of motion (ROM), bicep muscle strength (strength tests), and manual dexterity (BBT) were measured.	An EMG-VR hybrid system (QEMG-4XL) was used.	The intervention consisted of 20 sessions of biofeedback-boosted VR EMG, repeated twice a week (duration not specified). The methodology was based on VR games with functional strengthening exercises for reaching movements. The measurements were made before any intervention and again after each intervention.	Users with biofeedback-boosted VR showed significant improvements in elbow extension ROM, bicep strength, BBT, and maximum triceps muscle activity. However, it did not significantly improve coordination in three-dimensional movement acceleration.
17	The following variables were evaluated: isometric maximum strength of knee extensors, dorsiflexors, and plantarflexors of the ankle (PowerTrack II), balance (One-Leg Balance Test), walking speed (6MWT and 10MWT), and participation (Participation Assistance Scale).	The Wii Fit was used as a virtual reality system, and activities requiring weight shifting and movements, such as marching in place, squatting, and getting on and off the balance board, were selected for training. These activities were divided into two packs of four games (Balance Bubble, Table Tilt, Perfect 10, Super Hula Hoop, Ultimate Obstacle Course, Ski Jump, and Basic Step).	Participants underwent the intervention for 8 weeks, three times a week for 20 min, in addition to their usual therapy. The training consisted of playing 4 out of the 8 selected games, with each game played 12 times throughout the 24 sessions. Measurements were taken before the intervention and 8 weeks post-intervention.	99% of the sessions were completed, and improvements in muscle strength and walking speed were observed. Parents and children found the training understandable and not interfering with daily life, and there were no significant adverse events or severe falls.
18	Spasticity (MAS), grip strength (hydraulic dynamometer), pinch strength (pinch meter), and hand function (PDMS-2) were measured.	The Nintendo Wii was used as virtual reality material, and the following games were played for training: tennis, boxing, bowling, and basketball.	Participants were randomly divided into an experimental group (which received Wii training along with usual therapy) and a control group (which only received usual therapy). The experimental group underwent Wii training for 40 min, 3 times a week for 12 weeks. Measurements were taken at the beginning of the study and after the intervention.	The experimental group showed a reduction of 0.4 points in spasticity (MAS), an increase in grip strength of 1.6 kg in power grip and 1.2 kg in pinch grip, and hand function improved by 6 points on the PDMS-2 scale.
19	Anticipatory and reactive mechanisms of postural control were investigated in children and youth with CP. The tools used included a swaying platform, AI, cross-correlations, EMG, 6MWT, and the GMFM-CM.	The IREX was used for VR-based training, consisting of a 32-inch screen, a computer, a video camera, and a green screen for computer-generated images.	Participants were divided into an intervention group (N = 5) and a control group (N = 6). The intervention group performed VR-based balance training for 60 min daily over 5 days. Measurements were taken the weekend before and after the intervention, with the control group measured one week apart.	No significant differences in postural control were observed between the intervention and control groups.
20	The Assisting Hand Assessment (AHA) was used as the primary outcome measure, while the Melbourne Assessment of Unilateral Hand Function (MUUL) and the Canadian Occupational Performance Measure (COPM) were secondary measures.	The Hocoma Armeo Spring Pediatric, an exoskeleton combined with VR games, was used, along with a 1/16-inch aquaplast splint to restrict the unaffected limb during therapy.	The intervention took place over 2 weeks, with a total of 10 days and 6 h per day. The children wore the splint for 5 h and 30 min, using the exoskeleton for 30 min daily, and the remaining 30 min were dedicated to bimanual practice to incorporate newly learned unilateral skills into daily bimanual tasks. Measurements were taken before starting the intervention, immediately after, and 6 months after the intervention.	There was a significant improvement in bimanual performance (AHA) and satisfaction with performance (COPM), with a statistically significant improvement in unilateral function (MUUL), though not considered significant. Long-term effects persisted for 6 months after the intervention.
21	Gross motor function (GMFM-88) and spasticity level (MAS), static balance (SBT), muscle strength (LSUT and CUT), aerobic endurance (6MWT), and lower limb flexibility (SRT and MTT) were measured.	The virtual training used the Xbox 360 Kinect console, while traditional training used regular materials.	The intervention program involved golf training for 12 weeks, with 1 h sessions three days a week. In the first 2 weeks, basic golf movements were taught, and for the next 10 weeks, the group was split into two: one performed traditional golf training, and the other completed virtual golf training. Measurements were taken before and after the intervention.	Both training methods showed improvements in flexibility, muscle strength, aerobic endurance, and gross motor function, with no significant differences between the two types of training except for balance and lateral stepping tests.
22	Balance (PBS and Mini-BESTest), gross motor performance (GMFM-88), and daily functions (WeeFIM) were measured.	The Xbox 360 Kinect was used to play VR games, including Super Saver, Soccer, Volleyball, 20,000 Leaks, and Space Pop.	The intervention aimed to train balance and mobility, and participants were divided into two groups: the experimental group, which conducted VR games combined with physical therapy, and the control group, which only received physical therapy. The sessions lasted 60 min, 4 days a week for 6 weeks. Measurements were taken before, after, and 2 months after the 6-week intervention.	Measurements were taken before, 6 weeks after the intervention, and 2 months after the intervention. A significant improvement was observed in the Mini-BESTest, but there was no significant difference in motor performance or daily function between the experimental and control groups. The benefits were maintained after the 2-month follow-up.
23	Two measurements were taken: one before the intervention and another afterward, assessing gross motor function (GMFCS and GMFM).	A conventional exercise program and another using VR games with an Xbox 360 Kinect (Kinect Sports I, Kinect Joy Ride, and Kinect Adventures) were conducted.	Children were randomly divided into 2 groups of 15 (control and study groups). The control group completed a conventional exercise program for 60 min, 3 sessions per week for 3 months, while the study group performed the same conventional program along with 30 min of VR games per session. Measurements were taken before and after the intervention.	There were no significant differences in initial scores between the two groups. After the intervention, the study group showed a significant improvement in GMFM scores. Regarding GMFCS, there were no significant changes in either group, although the study group showed a significant difference between pre- and post-study scores.
24	The primary variable for hand function in bimanual activities was AHA, and secondary measures included BBT, JTTHF, and MFPT. The 6MWT gait evaluation was also performed. Questionnaires reported by parents included ABILHAND-Kids, ABILOCO-Kids, and ACTIVLIM-CP, with functional objectives measured by COPM.	A virtual REAtouch^®^ device was used, designed to facilitate therapeutic decision-making and structure the intervention to apply motor skill learning principles. It features a 45-inch responsive screen with a frame that allows height and tilt angle adjustments depending on the task at hand. Participants interacted using “bases” or direct contact with their hands or tangible objects that matched the target bases.	Participants were assigned to either a control group, which followed the “usual” HABIT-ILE protocols, or a study group, which underwent the same HABIT-ILE but with the REAtouch^®^ device used for half of the individual therapeutic time (41%). The intervention took place in a high-intensity day camp setting over 10–12 consecutive weekdays, totaling up to 90 h. Measurements were taken before, after, and 3 months after the intervention.	Both groups showed significant improvements in most outcome measures. The REAtouch^®^ group was not inferior to the HABIT-ILE group in motor function and functional goals. Significant changes in questionnaires indicated the transfer of learning to daily life activities. The integration of the REAtouch^®^ device into the HABIT-ILE camp was feasible and effective.

10MWT: 10-Meter Walk Test; 10ST: 10 Steps Climbing Test; 2MWT: 2-Minute Walk Test; 6MWT: 6-Minute Walk Test; ABILHAND-Kids: Manual Ability Measure for Kids; ABILOCO-Kids: Locomotion Ability Measure for Kids; ABMD: Areal Bone Mineral Density; AHA: Assisting Hand Assessment; BBT: Box and Block Test; BOTMP: Bruininks–Oseretsky Test of Motor Proficiency; BOTMP-2: Bruininks–Oseretsky Test of Motor Proficiency second edition; CB and M: Community Balance and Mobility Scale; CIMT: constraint-induced movement therapy; COPM: Canadian Occupational Performance Measure; CUT: Curl Up Test; DRT: Discriminative Reaction Time; DXA: Dual X-ray Absorptiometry; EHC: eye–hand coordination; EMG: Electromyography; FFRT: Functional Forward Reach Test; FSRT: Functional Sideways Reach Test; GHQ: General Health Questionnaire; GMFCS: Gross Motor Function Classification System; GMFM: Gross Motor Function Measure; GMFM-66: Gross Motor Function Measure-66; GMFM-88: Gross Motor Function Measure-88; GMFM-CM: Gross Motor Function Measure Challenge Module; GMFM-E: Gross Motor Function Measure-E; HABIT-ILE: Hand–Arm Bimanual Intensive Therapy Including Lower Extremities; HVCT: Home-Based Virtual Cycling Training; IREX: Interactive Rehabilitation and Exercise Systems; JTTHF: Jebsen–Taylor Test of Hand Function; LSUT: Lateral Step Up Test; MAUULF: Melbourne Assessment of Unilateral Upper Limb Function; MAS: Modified Ashworth Scale; MFPT: Manual Form Perception Test; Mini-BESTest: Mini-Balance Evaluation Systems Test; MTT: Modified Thomas Test; MUUL: Melbourne Assessment of Uni-lateral Hand Function; NDT: Neuro-Developmental Treatment; NPT: Nine-hole Peg Test; P-CIMT: Pediatric Constraint Induced Movement Therapy; PBS: Pediatric Balance Scale; PDMS-2: Peabody Developmental Motor Scale-2; PMAL: Pediatric Motor Activity Log; QOM: quality of movement; RCT: randomized controlled trial; REATouch^®^: Augmented Reality and Tactile Technology; ROM: range of motion; RSA: Running Speed and Agility; SBT: Static Balance Test; SD: Speed and Dexterity; SRT: Simple Reaction Time; STST: Sit-To-Stand Test; TGGT: Timed Get Up and Go Test; TUDS: Timed Up and Down Stairs; VMS: visual–motor speed; VRTT: Virtual Reality Treadmill Training; WeeFIM: Functional Independence Measure for Children.

## Data Availability

Not applicable.

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
