# Peer review of "Physical Exercise Interventions Using Virtual Reality in Children and Adolescents with Cerebral Palsy: Systematic Review"

_healthcare, 2025, doi:10.3390/healthcare13020189_

Round 1
Reviewer 1 Report
Comments and Suggestions for Authors
This systematic review demonstrated an interesting and important topic about the physical activity interventions using Virtual Reality in children and adolescents with Cerebral Palsy. However, from my perspective, there are some considerations that could improve the quality of this work.
# Background/Objectives
1. page 1, lines 21-23, please include the exact dates for the database search.
2. page 1, line 24, please specify the study type for the “24 studies eligible for inclusion”.
# Introduction
3. page 1, line 37, Correct the symbol in “per 1.000 live births” to a comma.
4. page 2, lines 71-76, what research gap you aimed to address, or what makes your review different from other similar reviews?
# Materials and Methods
5. Combine the sections “PICO Strategy” and “Inclusion and Exclusion Criteria” into one, as they contain overlapping content. Ensure the inclusion and exclusion criteria are clearly defined following the PICOS format (Population, Intervention, Comparison, Outcomes, and Study Design). Additionally, mention other criteria such as language in a separate study selection section. Further, I did not find information about the type of study you aim to include in the method section.
6. Including a PRISMA checklist in the Appendix might compliment your manuscript.
7. Table 1, please double check your criteria for population, the information is not consistent with Table 5, you aimed to included people aged 5-17 years, while the publications by Brien & Sveistrup, Tarakci et al., and Saussez et al. included individuals aged 18 years.
8. Table 1, clarify why your criterion for the outcome is “Provides benefits related to quality of life and willing to engage in PE”, why you only included positive outcome not negative outcome? What’s the primary outcome and secondary outcomes? This criterion appears inconsistent with your discussion stating “Most studies show greater benefits of VR compared to conventional training or therapy; however, in some cases, this difference is not seen [36,37], or it is shown in only half or fewer of the study variables”.
9. Search Strategy section, I wonder if the authors did a search in grey literature, it is desirable to search relevant grey literature sources, as suggested by the Cochrane Handbook
10. Search Strategy section, page 3, lines 120-121, this sentence should be included in results section not the method section.
11. Search Strategy section, page 3, lines 121-127, these should be included in a separate study selection section per the PRISMA checklist.
12. page 4, line 143, provide a reference for the RoB assessment.
13. Table 3-5, ensure that the year of publication is included in Tables 4-5, as stated.
14. There’s no information about synthesis methods per the PRISMA checklist, describe any methods used to synthesize results and provide a rationale for the choice(s).
# Results # Discussion
15. Provide a more detailed summary of study selection and study characteristics.
16. Move the description of results from the discussion section to the results section, e.g., page 20, lines 214-218 & 249-252.
Author Response
Thank you very much for your attention and review. We deeply appreciate the time you have taken to help us improve our article. All the changes based on your comments and suggestions have been carefully considered and incorporated into the revised version of the manuscript. To facilitate your review, the changes have been highlighted in green for ease of reference.
# Background/Objectives
Comment 1: Page 1, lines 21-23, please include the exact dates for the database search.
Response 1:
Thank you for your comment. The exact date of the search has been included has follows:
Methods: A systematic review was conducted and reported based on the PRISMA (Preferred Reporting Items for Systematic Review and Meta-analyses) statement. The search was conducted through the Web of Science, PubMed, and Scopus, databases on 1st September 2024. Studies based on PA interventions using VR in children and adolescent with CP were selected.
Comment 2: Page 1, line 24, please specify the study type for the “24 studies eligible for inclusion”.
Response 2:
Thank you for your comment. The type of studies has been specified as “experimental research articles”:
Results: A total of 24 experimental research articles were selected for this review. The studies included comprise a total sample on 616 participants between 4 and 18 years old.
# Introduction
Comment 3: Page 1, line 37, Correct the symbol in “per 1.000 live births” to a comma.
Response 3:
Thank you for your observation. The symbol has been corrected, replacing the period with a comma.
Comment 4: Page 2, lines 71-76, what research gap you aimed to address, or what makes your review different from other similar reviews?
Response 4:
Thank you for your comment. Our review aims to address a significant gap in research by adopting a comprehensive approach to physical exercise using virtual reality specifically in children and adolescents with CP. Unlike other reviews that focus solely on improving skills like balance or gait, or on rehabilitation in broader populations, this review examines the holistic application of virtual reality in promoting physical exercise. It explores how VR enhances physical activity adherence, motor skills, engagement, and other physical benefits. This perspective provides a broader understanding of VR's potential to improve health outcomes in this population.
# Materials and Methods
Comment 5: Combine the sections “PICO Strategy” and “Inclusion and Exclusion Criteria” into one, as they contain overlapping content. Ensure the inclusion and exclusion criteria are clearly defined following the PICOS format (Population, Intervention, Comparison, Outcomes, and Study Design). Additionally, mention other criteria such as language in a separate study selection section.Further, I did not find information about the type of study you aim to include in the method section.
Response 5:
Thank you for your comment. The two sections have been merged, and exclusion criteria now clarify the exclusion of non-experimental studies, case studies, and preliminary research. Inclusion criteria have also been updated to specify the study types. Additionally, the language used for the search (English and Spanish) has been mentioned under “Search Strategy.”
Comment 6: Including a PRISMA checklist in the Appendix might compliment your manuscript.
Response 6:
Thank you for your recommendation. A PRISMA checklist has been included in the appendix to complement the manuscript.
Comment 7: Table 1, please double check your criteria for population, the information is not consistent with Table 5, you aimed to included people aged 5-17 years, whilethe publications byBrien & Sveistrup, Tarakci et al., and Saussez et al. included individuals aged 18 years.
Response 7:
Thank you for pointing this out. The inclusion criteria and Table 1 have been revised to include participants aged up to 18 years to align with studies such as those by Brien & Sveistrup, Tarakci et al., and Saussez et al.
Comment 8: Table 1, clarify why your criterion for the outcome is “Provides benefits related to quality of life and willing to engage in PE”, why you only included positive outcome not negative outcome? What’s the primary outcome and secondary outcomes? This criterion appears inconsistent with your discussion stating “Most studies show greater benefits of VR compared to conventional training or therapy; however, in some cases, this difference is not seen [36,37], or it is shown in only half or fewer of the study variables”.
Response 8:
Thank you for your feedback. The “Outcomes” section in the PICO table has been revised for clarity. While the focus remains on outcomes of interest (e.g., improved quality of life and engagement in PE), we acknowledge the need to analyze both positive and negative results. The revised text highlights that although VR shows more benefits than conventional therapy in most studies, some studies report comparable or lesser outcomes. This does not negate benefits but reflects equivalent efficacy compared to standard approaches.
Comment 9: Search Strategy section, I wonder if the authors did a search in grey literature, it is desirable to search relevant grey literature sources, as suggested by the Cochrane Handbook
Response 9:
Thank you for your feedback, The authors opted to exclude grey literature from the PRISMA review due to several considerations. Firstly, grey literature often lacks the rigorous methodological standards and peer-review processes inherent to traditional academic publications. Secondly, the accessibility and retrievability of grey literature can be challenging, as it may not be indexed in standard databases. While acknowledging the potential value of grey literature, the authors prioritized the inclusion of quality, peer-reviewed studies to ensure the reliability and validity of the review. Furthermore, it makes easier to replicate the search methods.
Comment 10: Search Strategy section, page 3, lines 120-121, this sentence should be included in results section not the method section.
Response 10:
These lines have been relocated to the Results section as suggested.
Comment 11: Search Strategy section, page 3, lines 121-127, these should be included in a separate study selection section per thePRISMA checklist.
Response 11:
These lines have been moved to a newly created Study Selection section “Study registration procedure” to comply with the PRISMA checklist.
Comment 12: Page 4, line 143, provide a reference for the RoB assessment.
Response 12:
A reference for the Risk of Bias (RoB) assessment has been added to the manuscript.
Comment 13: Table 3-5, ensure that the year of publication is included in Tables 4-5, as stated.
14.
Response 13:
The publication year has been re-added to Tables 4 and 5, resolving the issue caused by the reference management software.
Comment 14: There’s no information about synthesis methodsper thePRISMA checklist, describe any methods used to synthesize results and provide a rationale for the choice(s).
Response 14:
Thank you for your comment. A paragraph describing the synthesis methods has been added to the Methods section::
Synthesis methods
The synthesis of results in this systematic review followed a narrative and descriptive approach, primarily integrating the findings from 24 studies. The narrative synthesis method was guided by the variability in study designs, interventions, and outcomes observed across the selected studies, which limited the feasibility of a meta-analytical approach. Furthermore, the data from eligible studies were categorized into general and specific variables, what facilitated the comparison and identification of recurring themes and patterns. Given the heterogeneity of interventions (use of various VR platforms like Nintendo Wii Fit, Xbox Kinect, and REAtouch®) and outcome measures (GMFM, muscle strength, balance, and engagement), a narrative synthesis allowed for a broader integration of qualitative and quantitative findings while acknowledging their contextual differences. Finally, results were presented in both tabular (Tables 5 and 6) and textual formats, highlighting the intervention impacts, such as improved motor functions, balance, and engagement levels, across diverse settings and VR systems​
# Results # Discussion
Comment 15: Provide a more detailed summary of study selection and study characteristics.
Response 15:
Thank you for your comment. Given the journal's word limit constraints, it is challenging to provide a more detailed summary without exceeding the allowed length. We have aimed to balance the level of detail with conciseness to meet the journal's requirements while ensuring that critical information is thoroughly presented. However, if further clarification is required on specific aspects, we would be happy to address these points in a revised version or provide additional information in supplementary materials, if allowed by the journal. Thank you for understanding our limitations.
Comment 16: Move the description of results from the discussion section to the results section, e.g., page 20, lines 214-218 & 249-252.
Response 16:
The identified lines (page 20, lines 214-218 and 249-252) have been moved to the Results section. Additionally, the Discussion section has been revised to summarize these points concisely.
Reviewer 2 Report
Comments and Suggestions for Authors
Context
The article entitled ‘Physical Exercise Interventions using Virtual Reality in Children and Adolescents with Cerebral Palsy: PRISMA Systematic Review’ deals with the rehabilitation of children and adolescents with cerebral palsy using virtual reality. The authors carried out a review over several years to list all the techniques.
Scientific contribution
The authors have carried out a systematic review of all cerebral palsy rehabilitation techniques using virtual reality. This article can help experts in the field of cerebral palsy, such as physiotherapists, to choose the best rehabilitation using virtual reality. What's more, virtual reality is fun for children and teenagers. It makes rehabilitation easier.
Suggestions for improvement
Introduction: the type of cerebral palsy used in the systematic review should be specified.
Method: why didn't you add the word ‘rehabilitation’ to the search for articles?
Discussion: the same systematic review should be carried out for other disabilities to see if it is possible to find a rehabilitation method using virtual reality for all types of disability.
Conclusion
In view of the scientific contribution of this article and its quality, I recommend acceptance.
Author Response
Thank you for your invaluable and positive feedback. We greatly appreciate your kind words and the time you have taken to review our work.
Comment 1: Introduction: the type of cerebral palsy used in the systematic review should be specified.
Response 1:
In the Introduction, we have added a short paragraph, highlighted in purple in the revised version of the article, specifying the types of CP. Furthermore, due to the limited number of articles addressing this specific topic, we have included all types of CP:
According to the National Institute of Neurological Disorders and Stroke (NINDS) [52], spastic CP, the most common type, is characterized by increased muscle stiffness and awkward movements, often affecting specific body parts. Hemiplegic CP impacts one side of the body, such as an arm and leg on the same side. Ataxic CP primarily affects balance and coordination, leading to shaky movements and difficulty with precise tasks. Dyskinetic CP involves uncontrolled, involuntary movements, including twisting and repetitive motions. Finally, mixed CP combines symptoms from more than one type, reflecting a combination of motor impairments.
Comment 2: Method: why didn't you add the word ‘rehabilitation’ to the search for articles?
Response 2:
We chose not to use the term “rehabilitation” in the article because our focus was on physical activity and exercise as an alternative to traditional rehabilitation for individuals with CP. Children with disabilities often primarily attend rehabilitation sessions, potentially missing out on other types of activities that offer additional benefits and promote long-term adherence. Our aim was to encourage interventions through physical exercise rather than framing them solely as rehabilitation.
Comment 3: Discussion: the same systematic review should be carried out for other disabilities to see if it is possible to find a rehabilitation method using virtual reality for all types of disability.
Response 3:
Thank you as well for your insightful comment regarding the potential of VR-based rehabilitation for other disabilities. We agree that this is a promising area for future research. Studies could investigate the efficacy of VR interventions for conditions such as spinal cord injuries or intellectual disabilities. A systematic review of such studies could provide valuable insights into the broader applicability of VR-based interventions across various neurological disorders.
Once again, thank you for your positive assessment of our article. We are grateful for the opportunity to contribute to this field and hope that our findings will inspire further research and innovative applications of VR technology.
Reviewer 3 Report
Comments and Suggestions for Authors
I believe it is not appropriate to mention PRISMA in the title of your manuscript.
It is not necessary to mention your search keywords in the abstract. Please add them as a supplementary file.
The result section in the abstract merely focused on the publication features rather than the actual results from review. Please revise.
In the method section there are many redundant sentences.
The result section is mainly a description of the included studies in the tables. It should be an analysis of the content you extracted rather than mentioning it in the tables.
Author Response
Thank you very much for your attention and review. We deeply appreciate the time you have taken to help us improve our article. All the changes based on your comments and suggestions have been carefully considered and incorporated into the revised version of the manuscript. To facilitate your review, the changes have been highlighted in blue for ease of reference.
Comment 1: I believe it is not appropriate to mention PRISMA in the title of your manuscript.
Response 1: The term "PRISMA" has been removed from the title, which now reads: "Physical Exercise Interventions Using Virtual Reality in Children and Adolescents with Cerebral Palsy: Systematic Review."
Comment 2: It is not necessary to mention your search keywords in the abstract. Please add them as a supplementary file.
Comment 3: The result section in the abstract merely focused on the publication features rather than the actual results from review. Please revise.
Response to comment 2 and 3:
Regarding your comments on the Methods and Results sections of the abstract, we have updated them as follows:
Methods: A systematic review was conducted and reported based on the PRISMA (Preferred Reporting Items for Systematic Reviews and Meta-Analyses) statement. The search was performed using the Web of Science, PubMed, and Scopus databases. Studies focusing on physical activity interventions using virtual reality (VR) in children and adolescents with cerebral palsy (CP) were selected.
Results: A total of 24 studies were included in this review, comprising a total sample of 616 participants aged between 4 and 18 years. The interventions ranged from brief sessions to intensive training programs. Results consistently demonstrated improvements in motor control, muscle strength, aerobic capacity, and overall participation in daily activities.
Comment 4: In the method section there are many redundant sentences.
Response 4: Thank you for your feedback, the Methods section has been revised to remove redundant sentences and ensure greater conciseness. For example, in the study and design section, we have summarized two sentences in one: "A systematic literature review was conducted using the PRISMA method to identify and select relevant scientific studies [20]."
Comment 5: The result section is mainly a description of the included studies in the tables. It should be an analysis of the content you extracted rather than mentioning it in the tables.
Response 5:
Thank you for your feedback on the Results section. Given the restricted word count of the article, we chose to present a concise Results section that primarily describes the included studies, as the remaining details are fully outlined in the tables. This approach minimizes redundancy by avoiding repetition of information already provided in tabular format. Nevertheless, the following paragraph have been added to the results section:
"
The studies included in this review comprise a total sample of 616 participants (602 of them with CP), ranging from 4 to 18 years old, with the most common age group being between 6 and 12 years. In terms of gender, it is specified for 407 subjects, of whom 228 are boys and 179 are girls, showing gender parity among participants. Likewise, the main inclusion criteria mentioned in the reviewed studies are primarily gross motor function levels I to III on the GMFCS scale, along with the ability to follow simple instructions. Less frequently, but also used in several studies, were GMFCS levels I to III, a MAS score below 2, and the MUUL scale, the latter only in the study by Winkels et al.(36). Regarding the types of CP studied, there is a wide heterogeneity in the sample, as several studies do not specify the type of CP (30,31,37,48). Others only mention that spastic CP is studied (32,33,41,45), with hemiplegic CP being the most mentioned type (34,35,38,40,47,49,50,53). On the other hand, ataxic CP is represented by a single subject (36), and dyskinetic CP by four subjects (42), among other types.
The studies analyzed different interventions programs using VR in children with CP across diverse goals, including improving motor function, balance, strength, hand-eye coordination, and overall functional mobility. VR platforms such as Nintendo Wii Fit, Wii Sports Resort, IREX, and custom systems were employed, with sessions ranging from short-term intensive programs to home-based interventions. Furthermore, Table 6 provides detailed and relevant information about the studies, including the variables and assessment tools used, the materials and types of VR employed, as well as the duration and type of exposure. It also outlines the specific measurement moments for the evaluated variables, offering a comprehensive overview of the methodologies and tools applied in the interventions. The duration and type of intervention have also been highly varied, ranging from a single session (30) to 12 weeks (32,33,42,47,50), and from mixed therapies dividing intervention time between some methodology and VR (30,38–42,45,47,51–53) to intensive PE training through VR lasting up to 5 and a half hours (49).
Among the most used types of VR in this review are the Wii (34,36,38,39,41–43,46,47), the Xbox Kinect (40,44,50–52) and the Interactive Rehabilitation and Exercise Systems - IREX (30,31,37,48)."
Once again, thank you for your invaluable feedback. We greatly appreciate your time and hope that the revised version of the manuscript addresses your comments and suggestions effectively.